# Animal Microbiomes as a Source of Novel Antibiotic-Producing Strains

**DOI:** 10.3390/ijms25010537

**Published:** 2023-12-30

**Authors:** Margarita N. Baranova, Ekaterina A. Pilipenko, Alexander G. Gabibov, Stanislav S. Terekhov, Ivan V. Smirnov

**Affiliations:** 1Shemyakin-Ovchinnikov Institute of Bioorganic Chemistry of the Russian Academy of Sciences, 117997 Moscow, Russia; baranova@ibch.ru (M.N.B.); gabibov@gmail.com (A.G.G.); 2Department of Chemistry, Lomonosov Moscow State University, 119991 Moscow, Russia

**Keywords:** wild microbiomes, secondary metabolites, antibiotics, probiotics, animal microbiota

## Abstract

Natural compounds continue to serve as the most fruitful source of new antimicrobials. Analysis of bacterial genomes have revealed that the biosynthetic potential of antibiotic producers by far exceeds the number of already discovered structures. However, due to the repeated discovery of known substances, it has become necessary to change both approaches to the search for antibiotics and the sources of producer strains. The pressure of natural selection and the diversity of interactions in symbiotic communities make animal microbiomes promising sources of novel substances. Here, microorganisms associated with various animals were examined in terms of their antimicrobial agents. The application of alternative cultivation techniques, ultrahigh-throughput screening, and genomic analysis facilitated the investigation of compounds produced by unique representatives of the animal microbiota. We believe that new strategies of antipathogen defense will be discovered by precisely studying cell–cell and host–microbe interactions in microbiomes in the wild.

## 1. Introduction

Known compound rediscovery is the major obstacle in the search for novel antibiotics from the pool of natural products. However, secondary metabolites of various microorganisms are far from depleted as a source of active compounds [1,2]. Potential improvements of systematic approaches to antibiotic discovery include (i) the development of new cultivation methods [3,4], (ii) the mining of antibiotic biosynthetic gene clusters (BGCs) of secondary metabolites followed by the activation of silent biosynthetic pathways [5,6], (iii) the creation of platforms to increase screening throughput [7,8,9,10], and (iv) the usage of less studied sources of producer strains [11,12,13].

The versatility of metabolic pathways makes microbes a fundamental component of any ecosystem [14,15]. However, their ecology remains poorly studied because of the complexity and methodological limitations [14]. Environmental conditions, including osmotic pressure, pH, temperature, limited resources, and biotic interactions, shape ecological niches [16]. Depending on the environment, microorganisms apply several basic survival strategies: outgrowing competitor strains through adaptation, mutualistic cooperation, and suppression of competitor strains [15]. Thus, the variety of ecological niches provides a variety of phenotypes, including metabolic profiles [16]. Antimicrobial production is one of the most common mechanisms of this suppression. Substances toxic to the surrounding community provide the producer with a selective advantage in fighting for limited space, light, minerals, or nutrients [15]. Nowadays, the structures of bacterial secondary metabolites represent the finest result of interactions between different species [17].

The classical source of antibiotic-producing strains is soil samples. The main discoveries of the “golden era” of antibiotics are associated with soil streptomycetes [1]. Later, bacteria from marine sediments were turned into an important source of microbiota [18,19]. At the same time, marine invertebrates were considered to be rich sources of various medicinal compounds. For a number of compounds, long after their discovery, it was revealed that the real producers of active metabolites were symbiotic microorganisms [20,21].

Microbes inhabit the organs of animals and the surface of their bodies, and in some cases, specialized organs are formed in the host organisms to contain cultures of symbionts [22]. Despite the ubiquity of free-living forms of bacteria, fungi, and archaea, most associations of microorganisms with animals turn out to be established. Since their inception, multicellular organisms have evolved together with microbial communities. The hologenomic model of animal evolution is currently being considered, in which animals act as units of natural selection together with symbiotic microflora [17,22,23]. The host animal supplies the microbial community with nutrients and ensures the relative stability of the environmental conditions. Microorganisms compete with each other and influence the host. The microbiota participates in maintaining the metabolic and immune functions of the animal, providing additional protection against pathogen invasion [22].

This review focuses on antimicrobial agents discovered through the study of animal-associated microbial communities and the enormous potential of these microbiota to produce evolutionarily optimized active compounds (Figure 1).

## 2. Marine Invertebrates

Historically, most natural bioactive compounds were found in terrestrial sources. This is explained by their greater availability compared to aquatic ones. However, improvements in sampling methods have revealed the huge diversity of marine biotopes [24]. The microbiome of marine invertebrates represents a highly competitive environment. The bacteria, archaea, and fungi that inhabit it are limited in nutrients, space, and light. High environmental pressure promotes the development of chemical defense strategies, including antibiotic production [24].

Most mature sponges are sessile animals that feed by filtration. The large surface of their bodies is covered with channels and pores that facilitate the accumulation of bacteria and fungi from the environment [24]. Sponges are hosts for diverse groups of bacteria and fungi, including both intracellular and extracellular symbionts. Microbiota make up 35–60% of an animal’s weight. Evolutionarily, sponges needed protection against pathogens. Symbionts able to secrete antimicrobial substances colonized healthy animals, coevolved with hosts, and gained competitive advantages [24].

Actinobacteria are classically considered to be the richest source of secondary metabolites. Their potential is well known from soil samples. Adaptation to the specific environment has made actinobacteria from marine sponges physiologically and genetically distinct from terrestrial species [13,25]. *Streptomonospora* sp. PA3 isolated from a sponge from the Persian Gulf showed inhibitory activity against *P. aeruginosa* and *S. aureus* [25]. Despite the smaller number of secondary metabolite biosynthesis clusters in the genome compared to terrestrial species, a novel antibiotic was identified among the strain’s metabolites—persiamycin A [25]. A wide range of biologically active compounds, including manzamines active against *Mycobacterium tuberculosis* and parasitic protists, have been obtained from Indonesian sponges [5]. Strains producing manzamines were described only later; for example, in the sponge *Acanthostrongylophora ingens*, the production of manzamines was provided by the actinomycete *Micromonospora* sp. [26,27]. Eight new antifungal compounds, antimycins I–P, were discovered in the culture of a sponge-associated strain of *Streptomyces sp.* NBU3104 [28]. Antimycin I showed high inhibitory activity against *C. albicans* and the phytopathogenic fungi *Penicillium expansum*, *Penicillium citrinum*, and *Botrytis cinerea*, while six rare acetylated antimycins and deformylated antimycin O are of interest from their structural point of view [28]. The application of novel cultivation methods, such as ichip, provides novel sponge-associated strains and compounds [29]. A new *Alteromonas* strain was derived from *Xestospongia muta* by employing the isolation chip into the sponge. The strain exhibited activity against *Staphylococci* and *Enterococcus faecium* mediated by a novel N-acyltyrosine [29].

In some sponges, the main contribution to the antibiotic activity of the symbiont community is made not by actinobacteria but by representatives of the genera *Pseudovibrio*, *Vibrio*, *Bacillus*, and others [24,30,31,32,33]. An unusual example of the main active metabolite produced by a bacterium of different genera is kocurin, a new member of the thiazolyl peptide family. Kocurin was isolated from the symbiont *Kocuria palustris* of the sponge genus *Xestospongia*. This substance was active against *Staphylococcus aureus*, including methicillin-resistant *S. aureus* (MRSA), *E. faecium*, and *B. subtilis* [34,35]. In addition to *K. palustris*, kocurin was detected in isolates of the sponge symbionts *K. marina* and *Micrococcus yunnanensis* [34]. The beta-carboline alkaloid norharmane, previously discovered in land plants, was found to be produced by *Pseudoalteromonas piscicida*, a symbiont of the sponge *Hymeniacidon perlevis* [36].

The sponge-associated strain identified as *Pseudomonas* sp. (*Pseudomonas fulva* was the closest relative strain) produced three high-molecular-weight peptides and two first-discovered alpha-pyrones (I and II) [37]. The more active alpha-pyrone I inhibited the growth of *B. subtilis*, *S. aureus* (MRSA), *M. catarrhalis*, and *E. faecium*, selectively affecting the membrane transport function of Gram-positive bacteria [37]. *Bacillus* strains with antagonistic properties against human pathogens have been repeatedly isolated from sponges [30,31,32]. Moreover, in some cases, antimicrobial activity was mediated by new antibiotics. Previously unknown thiopeptides YM-266183 and YM-266184 were isolated from *Bacillus cereus*, which are bacteria from the sponge *Halichondria japonica* [30]. In other cases, the antibiotic properties were provided by a set of metabolites characteristic of terrestrial species. For example, *B. pumilus* and *B. subtilis* strains obtained from the sponge *Aplysina aerophoba* were able to inhibit the growth of *S. aureus*, *E. coli*, *C. albicans*, and *S. epidermidis*, and this activity was provided by surfactins, fengycins, iturins, and lanthipeptides, such as subtilin and mersacidin [31]. *Aplysina aerophoba* is also known for the red pigment heptyl prodigiosin, which has antibiotic activity against *S. aureus*. In the sponge microbiome, it was produced by the gamma-proteobacterium *Pseudovibrio denitrificans* [33].

Freshwater sponges could also be an interesting and yet insufficiently studied source of strains [35]. Metagenomic analysis revealed that freshwater sponge microbiota was specific to their hosts and shared a number of genomic similarities with marine-sponge-associated bacteria. These similarities included defensive protein genes. We propose that the microbiota of freshwater sponges participates in the host’s defense against infection and is a prominent source of antimicrobials [38].

Coral polyps [39], mollusks [40], and echinoderms [41] also represent rich sources of antimicrobials. Lobophorin K is a cytotoxic antibiotic from the coral polyp *Lophelia pertusa* showing moderate and selective activity against Gram-positive bacteria [39]. Three known and two new lobophorins, H and I, were produced by a strain of *Streptomyces* sp. 1053U.I.1a.3b associated with the mollusk *Lienardia totopotens* [40]. Lobophorin I effectively inhibited the growth of the *Mycobacterium tuberculosis* culture, providing more than threefold selectivity for antituberculosis activity compared to cytotoxicity. Lobophorin H is probably a precursor compound, which does not affect the *M. tuberculosis* growth [40]. The strain *Streptomyces cavourensis* SV 21 was obtained from the sea cucumber *Stichopus vastus*. *S. cavourensis* SV 21 was found to produce valinomycin, a known depsipeptide, and a new analogue of valinomycin, streptodepsipeptide SV21 [41].

Tunicates are another group of marine animals known to be rich sources of bioactive metabolites [42]. Most of these chordates are sessile filter feeders in their mature form. This ecological niche contributes to the accumulation of microbiota. Due to the close interaction between the host and symbiont bacteria in the case of ascidians, it often remains unknown which organism is the true producer of the biologically active compound. For approximately 8% of the known secondary metabolites obtained from ascidians, there is evidence of an entirely bacterial origin [42]. *Lissoclimum patella* is a colonial sea squirt of the family *Didemnidae*, which is widespread in the western Pacific Ocean. It is one of the best-studied examples of bacterial symbiont–host relationships among marine systems [42]. *L. patella* contains photosynthetic symbionts, such as the cyanobacterium *Prochloron didemni*. These organisms synthesize cyanobactins, which are highly modified ribosomally synthesized peptides. The cyanobactin biosynthetic pathway is characterized by high tolerance to changes in precursor molecule cassettes [42]. *P. didemni*, a free-living bacterium, comprises panmictic oceanic populations [42,43]. Representatives of phylogenetically distant populations of symbionts occur in a single *L. patella* organism [43]. However, the metabolome of *L. patella* symbionts depends on the subspecies of the host organism, not on the geographical location. Symbionts of phylogenetically related ascidians produce similar sets of secondary metabolites. Thus, ascidians are able to modulate the spectrum of secondary metabolites of symbiotic bacteria [43]. *L. patella* and *P. didemni* are a classic example of a symbiosis between ascidians and bacteria but not the only one. Ascidians are characterized by species-specific microbiomes with unique biosynthetic fingerprints [44]. In addition to cyanobactins, ascidian symbionts produce alkaloids, modified peptides, and polyketides, many of which are considered potential candidates for clinical medicine [45].

Among actively moving marine invertebrates, the Hawaiian bobtail squid, *Euptymna scolopes*, contains the bacterial symbiont *Leisingera* sp., which has a known role in the host life cycle [46]. Cephalopods are characterized by the presence of a microbial community in the nidamental gland, an organ of the female reproductive system. *Leisingera* sp. penetrates inside eggs and selectively inhibits the growth of cultures of the genus *Vibrio* through the secretion of an indigoidine pigment [46].

## 3. Terrestrial Invertebrates

### 3.1. Nematodes

Nematodes and symbiotic gammaproteobacteria are one of the best-studied animal–bacterium systems. Representatives of the genus *Xenorhabdus* form a close association with entomopathogenic nematodes of the family *Steinernematidae*. These bacteria produce a wide range of compounds that suppress the growth of other bacteria, fungi, and protozoa. *Xenorhabdus* strains also inhibit the development of insects and nematodes competing with the host [47]. A similar system consists of bacteria of the genus *Photorhabdus* and nematodes of the family *Heterorhabditidae* [48].

Biologically active substances produced by *Xenorhabdus* spp. include depsipeptides [49], xenocoumacins [50], and genus-specific antimicrobial peptides [51] as well as benzylidene acetone, indole derivatives, and bacteriocins, the most famous of which is xenorhabdicin [52]. The diversity of BGCs in *Xenorhabdus* genomes significantly exceeds known compounds, and the biosynthetic potential of these bacteria remains underestimated [47]. In the case of *Streptomyces*, alternative cultivation methods are used to induce secondary metabolite production. This approach could also be used to discover products of *Xenorhabdus* strains. For example, the detection of amicoumacins among the secondary metabolites of *Xenorhabdus bovienii* was made possible by the use of a medium simulating the content of amino acids in the circulating fluid of the wax moth [53]. *Photorhabdus luminescens* produces a number of bacteriocins, including photorhabdicins and luminescines [54], as well as the known antifungal compound 3,5-dihydroxy-4-isopropylstilbene [55]. From the *Photorhabdus khanii* strain HGB1456, a new antibiotic, darobactin, was obtained [56]. Darobactin is a modified heptapeptide and shows activity against Gram-negative bacteria, including multidrug-resistant (MDR) strains of *E. coli* and *K. pneumoniae*. Darobactin is effective against Gram-negative bacteria in a murine model of sepsis, making it a promising drug candidate [56].

### 3.2. Insects

Insects make up the most numerous class of animals. More than a million species have been described, and they make up just a fraction of the total diversity. Insects occupy a huge number of ecological niches, and different hosts provide different habitats for the microbiota. Insect symbionts can be found on the cuticle, in the digestive tract, or inside cells and tissues. In addition to symbionts, antimicrobials were isolated from insect metabolites, such as honey, and nests [57]. Most studies of the antimicrobial activity of microbiota have been carried out with *Hymenoptera* species. Particular attention is paid to ants that grow fungi. The microbiome of their garden nests selectively prevents the development of entomopathogenic bacteria and fungi [57].

As streptomycetes are traditionally considered to be the main producers of antibiotics, most studies have been devoted to this genus only. Analysis of metagenomic data established that more *Streptomyces* strains were found in insect microbiomes than in water and marine sediments, making them an important source of antibiotic-producing strains [58]. The streptomycete strain ISID311 isolated from the mycobiome of the fungi-farming ant *Cyphomyrmex* sp. produced a new antimycotic compound called cyphomycin [58]. Cyphomycin has strong inhibitory activity against *Escovopsis* sp. fungi, which are pathogenic for ants. That may indicate its ecological significance for host nest protection. Cyphomycin also has antifungal activity against human pathogens, such as triazole-resistant *Aspergillus fumigatus* 11628, echinocandin-resistant *Candida glabrata* 4720, and MDR strain *Candida auris* B11211. Cyphomycin was tested in vivo in a model of disseminated candidiasis in mice and is considered to be a promising antifungal drug [58]. New pentacyclic polyketides that inhibit the growth of *B. subtilis*, *S. aureus* (MRSA), and vancomycin-resistant *Enterococcus faecium* (VRE) were obtained from a strain of *Streptomyces formicae* associated with the African ant *Tetraponera penzigi* [59]. These compounds were more effective than the structurally related compounds, fasamycins, which were also found among the secondary metabolites of the producer strain. The BGC of formamycin was further identified, and its formation by horizontal gene transfer was suggested [56,57]. A new polyene polyketide, selvamicin, was discovered in actinobacteria of the genus *Pseudonocardia* associated with the nest of *Apterostigma* ants [60]. Selvamicin is similar to nystatin and amphotericin but differs from them in the presence of a second sugar, a truncated macrocyclic core, and the absence of carboxylate and ammonium groups. These differences provide different pharmacokinetic properties of the compound, and, presumably, selvamicin has a different mode of action [57,60]. Selvamicin exhibits activity against *C. albicans* and a number of other fungi [57,60].

Termites are also known for cultivating fungi in their nests. The *Streptomyces* sp. M56 strain was obtained from the nest of African termites, *Macrotermes natalensis* [61]. The strain was first shown to produce natalamycin A, a compound of the ansamycin family, which showed antifungal activity both against *Pseudoxylaria* sp. X802 and against termite-cultivated *Termitomyces* T112 [61]. Further metabolomic analysis of the same strain revealed the production of efomycin M and two novel compounds—elaiophylin derivatives named efomycins K and L. The high antifungal activity of the strain was mediated by the synergistic effect of simultaneously produced geldanamycins, efomycins, and elaiophylins [62]. Other new compounds isolated from a termite symbiont strain were ilicolinic acids C and D [61]. These substances were produced by the fungus *Neonectria discophora*, obtained from the termite mound of the insect species *Nasutitermes corniger*. Both compounds exhibited moderate activity against *E. coli* and moderate cytotoxicity but were significantly less active compared to a previously discovered closely related compound, ilylicicolinic acid A [57,63].

A number of antibiotics, including frontalamides A and B and mycangimycin, have been identified by studying the microbiota of the southern pine beetle (*Dendroctonus frontalis*) [57,64]. The southern pine beetle participates in a multilateral symbiosis with fungi. *Entomocorticium* sp. is required for feeding larvae and is necessary for beetles to survive. *Ophiostoma minus* is an *Entomocorticium* antagonist, replacing *Entomocorticium* and leading to the death of the host. The bacterial community participates in maintaining the balance of fungal cultures through the production of secondary metabolites. Among the products of streptomycetes, frontalamides are polycyclic tetramate macrolactams [64], and mycangimycin is a polyene peroxide [65]. These compounds inhibited the growth of *O. minus*. Mycangimycin was also active against human pathogenic *C. albicans* [64].

Actinobacteria are the most thoroughly studied, but microorganisms of other phyla associated with insects also have the ability to produce antimicrobial compounds [57,66,67,68]. A strain of *Serracia marcescens* was discovered in the microbiome of the malaria vector mosquito *Anopheles stephensi* that inhibited the growth of *Plasmodium falciparum* and *B. subtilis*. The activity was mediated by the production of stephensiolides, cyclic lipodepsipeptides representing a novel family [66]. The bacteria *Burkholderia gladioli* of the unculturable strain Lv-StB were found on the eggs and glands of female darkling beetles, *Lagria villosa* [67]. This strain produced the polyketide lagriamide isolated from beetle eggs. Lagriamide was shown to be active against the entomopathogenic fungus *Purpureocillium lilacinum* [67]. Two novel compounds acting on the cell wall called lenzimycins A and B were isolated from a bacterium of the genus *Brevibacillus* associated with the dung beetle *Onthophagus lenzii*. Both of the lenzimycins had the ability to inhibit the growth of the entomopathogenic strain of *Bacillus thuringiensis* as well as the human pathogens *E. faecium* and *E. faecalis* [68].

## 4. Fish

Researchers turned to vertebrates as sources of microbiota that produce antimicrobial substances relatively late. There are few reports of new compounds found in such strains at present. However, microbial communities in the guts of fish are estimated to be more diverse than those of mammals. The associated communities vary within animal species depending on environmental conditions, such as water salinity or host diet [69,70]. A fish organism represents a unique environment, and the diversity of strains in such biotopes differs significantly from that in the surrounding seawater [69]. Stomach and intestinal microbiomes of fish are unique communities that contain strains phylogenetically distinct from previously known culturable bacteria [71].

Putative obligate psychrophiles were found in the microbiota of fish living in temperate climates [71]. Vibriosis is one of the most common bacterial diseases of fish, both on farms and in the wild. Vibrios are characterized by the rapid growth of cultures. If these bacteria colonize the intestines, they quickly become the dominant population. Virulence factors secreted by vibrios cause necrosis of fish tissues, slow growth and body malformation, blindness, and mortality. Vibriosis is normally prevented by both the host immune system of the fish and its microbiota. The search for mechanisms of such protection can contribute to both the production of valuable probiotics for fish farms and the search for compounds active against Gram-negative bacteria.

Marine *Actinobacteria* strains exhibiting activity against both Gram-negative and Gram-positive bacteria were isolated by studying fish intestinal microbiota communities [71]. New species of the *Paraoerskovia* genus were subjected to detailed analysis. The secondary metabolite responsible for the activity was purified and identified as the lipid sebasthenoic acid. Sebasthenoic acid inhibited the growth of the Gram-positive bacteria *B. subtilis*, *S. aureus*, and *E. faecium* [71]. Several strains with proven probiotic activity for fish have currently been characterized. The *Enterococcus durans* F3 strain from the intestines of freshwater fish *Catla catla* demonstrated tolerance to bile acid and gastric juice. *E. durans* F3 had a bactericidal effect against *S. aureus*, *E. coli*, *P. aeruginosa*, and *Salmonella typhi*. The effect was mediated by enterocin A, a known bacteriocin with a molecular weight of approximately 6.5 kDa [72].

Bacterial strains obtained from fish intestinal microbiota are used in aquaculture. *Rummeliibacillus stabekisii* [73] and various *Bacillus* species, including *Bacillus velezensis*, *Bacillus aryabhattai* and *Bacillus mojavensis*, are considered to be probiotic [74,75]. The advantage of fish-derived strains is their ability to colonize the intestines of fish and provide a prolonged effect [74]. Probiotic strains inhibited the growth of *Staphylococcus*, *Aeromonas,* and *Streptococcus* species [73,74]. A common feature of probiotic *Bacillus* strains is the secretion of lipase, amylase, and protease enzymes as well as nonribosomal bioactive metabolites [74,75].

The analysis of rainbow trout microbiomes revealed that the greatest diversity of microorganisms was found on the skin of the fish, followed by olfactory, gill, and intestine microbiomes [76]. The composition of skin microbial communities was reported to be similar in different species of teleost fish, with *Proteobacteria* and *Bacteroidetes* being the most represented [76]. Bacteria penetrate the fish skin epithelium and localize next to goblet cells. These communities differ from the general skin microbiome and are characterized by the dominance of *Firmicutes* and *Actinobacteria* [76]. It is supposed that the skin of teleosts provides a more favorable environment for colonization compared to the skin of terrestrial vertebrates [76]. The colonization of fish skin is improved by living epithelial cells that are free from dead keratinized layers. The microbiota on fish skin provides host protection against pathogens. *Arthrobacter* spp. (*A. stackebrandtii* and *A. psychrolactophilus*) and *Psychrobacter maritimus* inhibited the growth of two different aquatic pathogenic fungi, *Saprolegnia australis* and *Mucor hiemalis* [76].

## 5. Amphibians and Reptiles

The host habitat largely determines the diversity of the microbiota colonizing amphibians and reptiles [77]. The composition of a microbiota differs greatly even between animals of closely related species inhabiting the same region [77]. Different species of frogs have been shown to have a diverse composition of skin microbiota. Producers of antimicrobial agents highly active against both Gram-positive and Gram-negative pathogens were identified among members of this unique community [77]. The diversity of the microbiota of amphibians and reptiles is formed under the strong pressure of innate immunity. The specific components of amphibian immunity include diverse antimicrobial peptides (AMPs), which exhibit high activity and are characteristic of these animals [78,79]. The specific microbiota of amphibians and reptiles often contains antibiotic-resistant and multidrug-resistant *Enterobacteriaceae* strains, representing a risk factor for pathogen transmission [80,81].

Bacteriocins were frequently identified in the microbiomes of amphibians and reptiles, including nisin Z produced by *Lactococcus lactis* [82,83]. Nisin Z inhibits the growth of the amphibian pathogens *Citrobacter freundii* and *Listeria monocytogenes*. Prodigiosin, violacein, and volatile organic compounds are produced by the skin microbiota of geographically distant amphibian species [84]. These metabolites inhibit the growth of pathogenic fungi of the genus *Batrachochytrium* that cause chytridiomycosis in amphibians [84]. The symbiotic strains of the bacteria *Serratia marcescens* produce prodigiosin. The antimicrobial effect of prodigiosin was mediated by intercellular contacts with the pathogen *Staphylococcus aureus* [85]. Strains of *Bacillus atrophaeus* were identified in the skin microbiota of the Iranian marsh frog (*Rana ridibunda*) [86]. *Bacillus atrophaeus* exhibited a wide range of antimicrobial and antiproliferative activities [86].

*Pseudomonas aeruginosa* strains were isolated from the Amboina box turtle (*Cuora amboinensis*) [87]. *Pseudomonas aeruginosa* demonstrated antibacterial activity against Gram-positive bacteria (*B. cereus*, *Streptococcus pyogenes*, and *Staphylococcus aureus* MRSA) and Gram-negative pathogens (*E. coli* K1, *S. marcescens*, *P. aeruginosa*, *S. enterica*, and *K. pneumoniae*). Novel N-acyl-homoserine lactones, 4-hydroxy-2-alkylquinolines, and rhamnolipids were identified as active metabolites of *P. aeruginosa* [87]. A similar profile of metabolites was shown for the microbiota of the saltwater crocodile (*Crocodylus porosus*) [88]. In this case, the activity was also associated with *Pseudomonas* strains.

## 6. Birds

Despite the wide distribution of birds, the diversity of the avian microbiome has been studied rather limitedly. This is largely due to the difficulties associated with the intravital selection of native biomaterials. Moreover, metatranscriptomic data indicate that antibiotics associated with environmental pollution have a high impact on biodiversity and antibiotic resistance in the bacterial communities of wild birds [89]. The antibiotic impact is particularly acute due to the large-scale use of antibiotics in poultry farming [90]. Probiotics provide a highly effective alternative to antibiotics in agriculture. The application of probiotic strains of *Enterococcus faecium*, *Pediococcus acidilactici*, *Bacillus animalis*, *Lactobacillus salivarius*, and *Lactobacillus reuteri* prevents the colonization of birds by pathogens [91]. However, it has been repeatedly shown that specific biodiversity and the stability of bacterial communities are extremely important for the normal physiology of birds [92]. There are numerous studies on the functionality of the chicken microbiota [93,94,95] despite limited knowledge of the antimicrobial potential of wild birds.

In most cases, it was reported that the molecular mechanisms underlying microbiome stability in birds are associated with the production of short-chain fatty acids by *Lactobacillus*, *Clostridium*, and *Streptococcus* strains [92,93,94]. Class II lanthipeptides, including FK22 [96], OR7 [97], L-1077 [98], and salivaricin SMXD51 [99], are the most studied antimicrobials produced by *Lactobacillus* [100]. Lactic acid bacteria (LAB) with potent antimicrobial activity were isolated from the microbiome of griffon vulture (*Gyps fulvus* subsp. *fulvus*) [101]. It was identified that LAB *Enterococcus faecium* M3K31 produces enterocin HF (EntHF), which is highly active against *Listeria* spp. [101]. Plumage [102] and coccygeal gland [103,104] microbiomes are also particularly important for birds.

*Proteobacteria*, *Actinobacteria*, and *Bacteroidetes* were reported to be the most abundant phyla in the passerine feather microbiota, while *Alphaproteobacteria*, *Gammaproteobacteria*, and *Betaproteobacteria* are the most abundant classes [102]. The exceptions were the sand swallow and the common redstart, which had a poorer microbiome of feathers and a predominance of representatives of the *Firmicutes* phylum and the *Bacilli* class. *Streptococcus* and *Lactobacillus* genera were the most represented. The number of potential bacteriocin producers was negatively correlated with the overall diversity of the feather microbial community. A positive correlation between the number of bacteria capable of destroying keratin and bacteriocin producers was detected [102]. Many *Bacillus* and *Pseudomonas* species have protective properties due to their ability to inhibit the growth of other microorganisms, although some of them are considered harmful to the host bird [102].

*Enterococcus faecalis* producing specific bacteriocins was isolated from the coccygeal gland of the hoopoe (*Upupa epops*) [104]. Hoopoes are characterized by a different composition of fatty secretion of the gland depending on the nesting phase. The liquid for feather lubrication becomes dark and foul smelling in nesting females and in chicks [103]. *Enterococcus faecalis* was found to be the dominant culture in the secretion of the coccygeal gland, and enterocins were identified as the main antimicrobials. The combination of enterocins MR10A and MR10B inhibited the growth of *S. aureus*, *B. cereus*, and *E. faecalis* strains, different from the producer strain [103]. A more recent study has shown that *Enterococcus faecalis* is the dominant strain in hoopoe coccygeal gland secretions regardless of the individual bird or nest, and strains with greater potential for producing enterocins are more common [104].

## 7. Mammals

Similarly to birds, the microbiome of wild mammals has been studied much more fragmentarily than the microbiome of farm animals. Another important similarity between the microbiome of birds and mammals is the high importance of LAB (*Lactobacillus* species, *Bifidobacterium* spp., and *Bacillus* spp.), which have antagonistic properties towards pathogens and are widely used as probiotics in animal husbandry [105]. *Ligilactobacillus salivarius* strains were isolated from the microbiome of calves [106]. *L. salivarius* exhibited antagonistic properties against pathogenic strains of *E. coli*, demonstrating a probiotic effect in a rat model in vivo. Streptococci are also widespread members of the bovine microbiome, well known for the production of lantipeptides of the bovicin family [107].

Marine mammals with global migration routes are exposed to dramatic changes in their environment. Migrating marine mammals have an increased diversity of microbiomes compared to nonmigratory species [108]. A strain of *Micromonospora auratingra* was isolated from the intestinal microbiota of harbor porpoise (*Phocoena phocoena*) [108]. The strain selectively inhibited the growth of Gram-positive bacteria, including the intestinal pathogen *Clostridium difficile*. The activity of *M. auratingra* was provided by a new glycosylated polyketide antibiotic called focoenamycin [108]. The antibiotic showed low cytotoxicity. Focoenamycin shares a number of common structural features with fidaxomicin, a known drug effective against *C. difficile*, but has a different mechanism of action [108].

The human microbiota is the most extensively studied object [109]. It exhibits a high number of bacteriocin-producing strains, which play an extremely important role in the ecology of the human microbiome [110,111,112,113]. A lantibiotic produced by the commensal bacterium *Blautia producta* prevented colonization by vancomycin-resistant strains of *E. faecium* (VRE), restoring the sensitivity of enterococci to vancomycin [114]. Bacteroidetocins represent class IIa Gram-positive bacteriocins, inhibiting the growth of *Bacteroides*, *Parabacteroides,* and *Prevotella* species [115].

Secondary bile acids represent a unique group of antimicrobial agents from the human microbiome. These compounds are products of human biosynthesis pathways modified by bacteria. A new form of lithocholic acid (isoallolithocholic acid) was demonstrated to have high activity against the pathogenic bacteria *Clostridioides difficile* and *Enterococcus faecium* [116]. Moreover, the human microbiota could serve as a source of classical antibiotics. Lugdunin is a product of nonribosomal peptide synthetases (NRPS) produced by the human microbiota commensal *Staphylococcus lugdunensis* [117]. The microbiota of wild animals could also be a source of antibiotics synthesized by polyketide synthase (PKS)/NRPS. It has been repeatedly observed for amicoumacin A, which is highly abundant in a variety of different animal species [118,119].

## 8. Discussion

The animal microbiome represents a unique reservoir for antibiotic discovery [77,110,113,118,119,120] (Table 1).

Probiotic microorganisms were repeatedly isolated from natural microbiota sources [121,122], representing an extensive reservoir for antimicrobial agent research. Most probiotic strains and commensal bacteria have an indirect effect on the microbiome by influencing the immune system of the host [123] or producing functionally important enzymes [124]. However, direct killing of pathogens is characteristic of the most ubiquitous commensals. In addition to bacteria with probiotic effects, pathogens are also able to produce antimicrobial agents, playing a fundamental role in the interaction of bacteria in vivo. The vast majority of *Pseudomonas aeruginosa* strains show antagonism towards *Staphylococcus aureus*, mediated by the synergistic effects [7,125] of pyocyanin [126], phenazine-1-carboxylic acid [127], and 2-heptyl-4-hydroxyquinoline [128]. The same bacteria may play diametrically opposite roles in different animal hosts. *Pseudomonas* are common symbionts of reptiles and amphibians. At the same time, *Pseudomonas* are pathogens of mammals.

In contrast, “universal probiotic commensals” have also been reported [119]. *Bacillus* species have been identified as the normal microbiota of a wide repertoire of organisms, including marine invertebrates, insects, and mammals. These strains often share a common “fingerprint” of secondary metabolites, of which the antibiotic amicoumacin is the most active antimicrobial agent. *Bacillus* species have a rich arsenal of highly active secondary metabolites [129]. *Bacillus* species not only play an important role as a common component of the microbiota of wild animals but are also widely used in practice as a probiotic for poultry farming [130], pig farming [131], aquaculture [132], and human healthcare [133].

The contaminated environment itself can be a source of antimicrobial agents [120]. However, microbiomes are highly sensitive to the “quality” of the environment [77]. It creates a high risk of degeneration of the natural biodiversity, leading to the loss of antibiotic producers as a result of destructive human activity.

Despite the enormous amount of data accumulated from host–cell interactions in the wild [134], we are still far from having a detailed understanding of the entire landscape of molecular interactions in microbiomes. Moreover, unique antimicrobial agents may be found right in front of our nose [117]. Modern technologies will undoubtedly make a major contribution to the identification of new antibiotics. Genome mining [6,135] and ultrahigh-throughput microfluidic technologies [9,10,118,136,137] will allow us to detail the unique biodiversity of antibacterial agents associated with wildlife microbiomes.

## Figures and Tables

**Figure 1 ijms-25-00537-f001:**
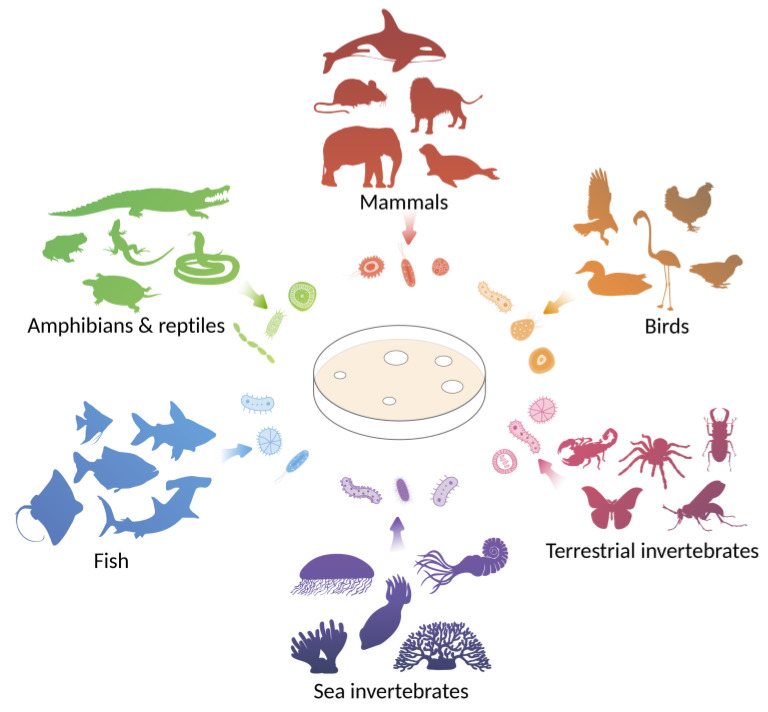
Animals are considered as sources of unique communities of associated microorganisms that exhibit antagonistic activity against pathogenic strains. Sources of microorganisms considered in the review are marine invertebrates, terrestrial invertebrates, fish, amphibians and reptiles, birds, and mammals.

**Table 1 ijms-25-00537-t001:** Examples of new antimicrobial compounds identified from animal-associated microorganisms.

Bacteria		Hosts	
**Sea Invertebrates**			
*Kocuria palustris*	Sponge	*Xestospongia* sp.	[34,35]
*Kocuria marina*			
*Micrococcus yunnanensis*			
*Micromonospora* sp.	Sponge	*Acanthostrongylophora ingens*	[26,27]
*Streptomyces* sp. NBU3104	Sponge		[28]
*Pseudomonas* sp.	Sponge		[37]
*Bacillus cereus*	Sponge	*Halichondria japonica*	[30]
*Streptomyces* sp. M-207	Cnidarian (coral polyp)	*Lophelia pertusa*	[39]
*Streptomyces* sp. 1053U.I.1a.3b	Mollusk	*Lienardia totopotens*	[40]
*Streptomyces cavourensis* SV 21	Echinoderm	*Stichopus vastus*	[23]
*Prochloron didemni*	Chordate (ascidian)	*Lissoclimum patella*	[42,43]
**Terrestrial Invertebrates**			
*Xenorhabdus doucetiae* DSM17909	Nematode	*Steinernema diaprepesi*	[49]
*Xenorhabdus nematophila*	Nematode	*Steinernema* sp.	[51]
*Xenorbabdus* sp. strain Q1, *Xenorbabdus nematopbilus* All	Nematode	*Steinernema* sp., *Steinernema filtiae*	[50]
*Photorhabdus khaini*	Nematode	*Heterorhabditis* sp.	[56]
*Streptomyces* sp. ISID311	Insect (ant)	*Cyphomyrmex* sp.	[58]
*Streptomyces formicae*	Insect (ant)	*Tetraponera penzigi*	[59]
*Pseudonocardia* sp.	Insect (ant)	*Apterostigma*	[60]
*Streptomyces* sp. M56	Insect (termite)	*Macrotermes natalensis*	[61]
			[62]
*Neonectria discophora*	Insect (termite)	*Nasutitermes corniger*	[63]
*Streptomyces* sp.	Insect (beetle)	*Dendroctonus frontalis*	[64]
*Streptomyces* sp.	Insect (beetle)	*Dendroctonus frontalis*	[65]
*Serracia marcescens*	Insect (fly)	*Anopheles stephensi*	[66]
*Burkholderia gladioli* Lv-StB	Insect (beetle)	*Largia villosa*	[67]
*Brevibacillus* sp.	Insect (beetle)	*Onthophagus lenzii*	[68]
**Fish**			
*Paraoerskovia* sp.	Fish (cod)	*Lotella rhacina*	[71]
**Amphibians and Reptiles**			
*Pseudomonas aeruginosa* CM3	Reptile (turtle)	*Cuora amboinensis*	[87]
**Birds**			
*Enterococcus faecium* M3K31	Bird (vulture)	*Gyps fulvus subsp. fulvus*	[101]
*Enterococcus faecalis*	Bird (hornbill)	*Upupa epops*	[103]
**Mammals**			
*Micromonospora auratingra*	Mammal (cetacean)	*Phocoena Phocoena*	[108]

## Data Availability

Data are contained within the article. Additional data are freely available on request from the corresponding author.

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
