# Peer review of "Animal Microbiomes as a Source of Novel Antibiotic-Producing Strains"

_ijms, 2023, doi:10.3390/ijms25010537_

Round 1
Reviewer 1 Report
Comments and Suggestions for Authors> Reviewer comment
General comments
Natural compounds remain the primary wellspring for discovering new antimicrobials, surpassing the count of already known structures. Yet, the recurrent identification of familiar substances prompted a shift in both approaches to antibiotic exploration and the origins of producing strains. The influence of natural selection and the intricate web of interactions within symbiotic communities positions them as promising founts for pioneering compounds. This discussion delves into microorganisms associated with diverse animals, specifically examining their antimicrobial properties. Implementing alternative cultivation methods, ultrahigh-throughput screening, and genomic analysis streamlines the exploration of compounds synthesized by distinct members of animal microbiota. Our conviction lies in the prospect of unveiling fresh anti-pathogen defense strategies through meticulous examination of cell-cell and host-microbe interactions within wild microbiomes.
General comment
Please make highlighted any new changes
Abbreviation, please indicate full words for first use at the text of article.
References should be based on journal constriction.
# Reviewer comment #
1- Abbreviation all over your paper should be cited correctly
2- References should be based on journal constriction.
# Reviewer comment #
3- Ensure accurate citation of abbreviations throughout the paper.
4- References need to strictly adhere to journal conventions.
5- Provide both general and specific comments within the manuscript.
6- Title requires improvement.
7- Introduction should clearly outline the problem.
8- Highlight any new changes made.
9- While there is a vast amount of data, a more comprehensive understanding of molecular interactions in microbiomes is necessary. Additionally, novel antimicrobial agents might be present in readily accessible sources. Advanced technologies, such as genome mining and ultrahigh-throughput microfluidic approaches, will significantly contribute to identifying new antibiotics and elucidating the diverse antibacterial agents within wildlife microbiomes.
Comments on the Quality of English LanguageMinor revision
Author Response
The authors of MS entitled “Animal microbiomes as a source of novel antibiotic-producing strains” express their deepest appreciation to the reviewers and thank them for their important comments.
The team of authors expresses special gratitude to Reviewer 1, whose extremely valuable comments enable us to significantly improve the MS as well as reveal some unobvious details.
We completely agree with the respective comments and prepared a detailed answer to each of them, including the modifications to the MS.
Specifically, for the comments of Reviewer 1:
General comment
Please make highlighted any new changes
Changes are highlighted in yellow.
Abbreviation, please indicate full words for first use at the text of article.
Abbreviations were checked to be placed when they were first mentioned.
Staphylococcus aureus, including methicillin-resistant S. aureus (MRSA)
vancomycin-resistant Enterococcus faecium (VRE)
polyketide synthase (PKS)
were introduced.
References should be based on journal constriction.
Refs were checked according to IJMS recommendations.
# Reviewer comment #
- Abbreviation all over your paper should be cited correctly
Abbreviations were checked to be placed when they were first mentioned.
Staphylococcus aureus, including methicillin-resistant S. aureus (MRSA), vancomycin-resistant Enterococcus faecium (VRE), and polyketide synthase (PKS), were introduced.
- References should be based on journal constriction.
According to Reviewer1’s suggestion, all the references were modified according to IJMS guidance.
- Ensure accurate citation of abbreviations throughout the paper.
Accurate citations of abbreviations were provided.
- References need to strictly adhere to journal conventions.
Refs were checked according to IJMS recommendations.
- Provide both general and specific comments within the manuscript.
According to Reviewer1, we added a small commentary in the introduction section.
The versatility of metabolic pathways makes microbes a fundamental component of any ecosystem [14, 15]. Their ecology remains poorly studied because of its complexity and methodological limitations [14]. Environmental conditions, including osmotic pressure, pH, temperature, limited resources, and biotic interactions shape ecological niches [16]. Depending on the environment, microorganisms apply several basic survival strategies: outgrowing competitor strains through adaptation, mutualistic cooperation, and suppression of competitor strains [15]. Thus, the variety of ecological niches provides a variety of phenotypes, including metabolic profiles [16]. Antimicrobial production is one of the most common mechanisms of this suppression. Substances toxic to the surrounding community provide the producer with a selective advantage in fighting for limited space, light, minerals, or nutrients [15]. Nowadays, the structures of bacterial secondary metabolites represent the finest result of interactions between different species [1].
- Title requires improvement.
We suggest the MS title strictly explains its main idea: to represent the biodiversity of ecological niches of animal microbiomes as a source of new antimicrobials.
- Introduction should clearly outline the problem.
We added the finalizing sentence to the introduction section to clearly outline the problem of the MS.
This review focuses on antimicrobial agents discovered through the study of animal-associated microbial communities and the enormous potential of these microbiota to produce evolutionarily optimized active compounds (Figure 1).
- Highlight any new changes made.
The changes made are highlighted in yellow.
This review focuses on antimicrobial agents discovered through the study of animal-associated microbial communities and the enormous potential of these microbiota to produce evolutionarily optimized active compounds (Figure 1).
- While there is a vast amount of data, a more comprehensive understanding of molecular interactions in microbiomes is necessary. Additionally, novel antimicrobial agents might be present in readily accessible sources. Advanced technologies, such as genome mining and ultrahigh-throughput microfluidic approaches, will significantly contribute to identifying new antibiotics and elucidating the diverse antibacterial agents within wildlife microbiomes.
We fully agree with Reviewer1.
Reviewer 2 Report
Comments and Suggestions for Authors
In this review article, the authors focus on the importance of the animal microbiome for the identification of new antibiotics and antimicrobial structures. The article covers the relevant literature, is clearly structured and well written. However, as a suggestion for improvement, I would like the authors to go into more detail on the importance of the ecological niche in general and the resulting assumption for the discovery of new antibiotics (especially considering ecological pressure).
Author Response
The authors of MS entitled “Animal microbiomes as a source of novel antibiotic-producing strains” express their deepest appreciation to the reviewers and thank them for their important comments.
The team of authors expresses special gratitude to Reviewer 2, whose extremely valuable comment enables us to reveal some unobvious details that significantly improve the MS.
We completely agree with the respective comment and have prepared a detailed answer.
Specifically, for the comment of Reviewer 2:
However, as a suggestion for improvement, I would like the authors to go into more detail on the importance of the ecological niche in general and the resulting assumption for the discovery of new antibiotics (especially considering ecological pressure).
According to the reasonable suggestion of Rewiever 2, detail on the importance of diversity of ecological niches and interactions in the microbial community for antibiotic production. We add respective information in the Introduction section:
The versatility of metabolic pathways makes microbes a fundamental component of any ecosystem [14, 15]. Their ecology remains poorly studied because of its complexity and methodological limitations [14]. Environmental conditions, including osmotic pressure, pH, temperature, limited resources, and biotic interactions shape ecological niches [16]. Depending on the environment, microorganisms apply several basic survival strategies: outgrowing competitor strains through adaptation, mutualistic cooperation, and suppression of competitor strains [15]. Thus, the variety of ecological niches provides a variety of phenotypes, including metabolic profiles [16]. Antimicrobial production is one of the most common mechanisms of this suppression. Substances toxic to the surrounding community provide the producer with a selective advantage in fighting for limited space, light, minerals, or nutrients [15]. Nowadays, the structures of bacterial secondary metabolites represent the finest result of interactions between different species [21].